# Adaptive Continual Learning Through Proactive Detection of Transfer and Interference

## Abstract

Continual learning (CL) requires models to sequentially learn multiple tasks, maximizing transfer and minimizing interference. However, current methods cannot proactively detect all types of transfer and interference at the local optimization level, limiting their effectiveness. To address this, we propose an adaptive continual learning strategy by proactively detecting transfer and interference. We derive the conditions for all types of transfer and interference from the perspective of parameter sharing and optimization, based on the Fisher matrix and gradient update directions. Using this, we proposed a task transfer distance metric to help model modules detect transfer and interference. We propose a dynamic parameter update mechanism and a dynamic expansion strategy, using inserted adapters in the pre-trained model, to manage all types of transfer and interference. Experiment results on seven benchmarks show that our method achieves the best accuracy with limited parameters, maximizing transfer and minimizing interference.

## 1 Introduction

Recently, artificial intelligence has made many significant breakthroughs across various fields. However, the traditional approach of repeatedly training models on fixed datasets has resulted in high costs and delayed updates. To address this challenge, some researchers have proposed continual learning (Masana et al., 2023). Continual learning enables models to learn from data streams in open, dynamic environments without access to previously encountered data (Masana et al., 2023). During this process, different types of transfer and interference can arise (Wang et al., 2024). Backward interference occurs when the model's performance declines on learned tasks during learning new tasks, named catastrophic forgetting (Wang et al., 2024). Forward interference happens when excessive protection of old knowledge prevents the model from effectively learning new ones (Zhou et al., 2024c). On the other hand, forward transfer occurs when old knowledge helps accelerate learning on new tasks, while backward transfer happens when learning a new task improves performance on earlier tasks (Wang et al., 2024; Masana et al., 2023). The primary goal of continual learning is to maximize transfer between tasks while minimizing interference (De Lange et al., 2022).

Some studies suggest that transfer and interference in continual learning are linked to the sharing and overlap of model parameters (Wang et al., 2024). Most current methods are designed to maximize transfer while minimizing interference Rebuffi et al. (2017); Kirkpatrick et al. (2017). Replay-based methods (Luo et al., 2024; Rebuffi et al., 2017) achieve it by storing, replaying, or generating samples from previous tasks, simulating repeated training on fixed datasets. However, as tasks increase, the required storage and computational resources increase uncontrollably, leading to issues like sample imbalance. Dynamic network-based methods (Bonato et al., 2024; Yoon et al., 2017; Wang et al., 2022a) promote forward transfer by reusing frozen old parameters while adding new ones to avoid interference with new tasks. However, freezing old parameters limits backward transfer, the network size grows uncontrollably with tasks added. optimization-based methods (Kao et al., 2021; Saha et al., 2021; Saha & Roy, 2023; Zeng et al., 2019) reduce backward interference and encourage forward transfer by preventing the overlap of important model parameters. These approaches hinder backward transfer and increase forward interference when new and old tasks share too many important parameters. Recently, many studies have integrated fine-tuning of Pre-Trained Models (PTMs) with these continual learning methods (Liang & Li, 2024; Yu et al., 2024; Qiao et al., 2023; Zhou et al., 2024b; Luo et al., 2024), demonstrating superior performance (Zhou et al., 2024a). However, the methods mentioned above do not provide a detailed analysis of the specific

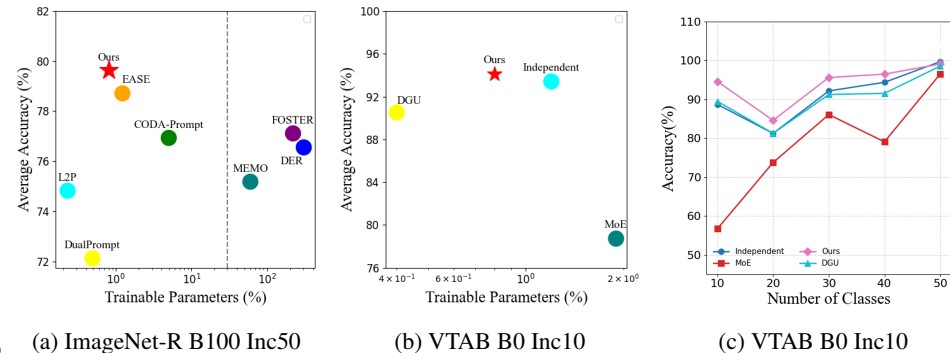

htbp      (a) ImageNet-R B100 Inc50          (b) VTAB B0 Inc10          (c) VTAB B0 Inc10

Figure 1: **Parameter-performance comparison. (a):** The comparison of different methods on ImageNet-R B100 Inc50. **(b):** The comparison of different variants on VTAB B0 Inc10. **(c):** Stage accuracy of different varaints on VTAB B0 Inc10.

conditions under which different types of transfer and interference occur. Instead, they generally avoid interference by preventing parameter overwriting and achieve transfer by freezing parameters, which fails to proactively detect and properly handle all types of transfer and interference.

To address this issue, we have made the following efforts. (1) **Theoretical derivation.** We first discovered through theoretical derivation that transfers and interference between tasks are related to the extent of parameter sharing and the optimization directions of shared parameters. Then, we defined the conditions under which different types of transfer and interference occur (see Section 2). (2) **Transfer distance metric.** Based on the conditions, we leveraged the powerful representational capability of the pre-trained ViT model to estimate the task Fisher matrix. By combining this with gradient update directions, we propose a transfer distance metric to quantify the degree of shared parameters and their optimization relationship, helping to identify transfer and interference (see Section 4.2). (3) **Adaptive continual learning strategies.** Using this metric, the model can actively detect transfer and interference during continual learning. We insert adapter modules at each layer to fine-tune the pre-trained model for new tasks. When tasks show high transfer or low relevance, they share the same adapter and apply dynamic gradient updates. This method adjusts the optimization trajectory based on transfer distance, maximizing transfer while minimizing interference. In cases of high interference, we introduce new adapters and activate the frozen old adapters to assist new task learning. Our adaptive continual learning method enables model modules to proactively detect transfer and interference and select appropriate continual learning strategies, achieving an optimal balance between accuracy and resource efficiency (see Section 4.3).

We validate our method on seven benchmarks. As shown in Fig. 1, our method achieves the best accuracy with limited parameters, effectively balancing accuracy and parameter efficiency. We also analyzed the method's parameter sharing and transfer across benchmarks, demonstrating its ability to detect transfer and interference while selecting strategies to maximize transfer and minimize interference. Ablation studies further confirm the effectiveness of each component.

## 2 BACKGROUND

Continual learning is learning from samples of different distributions $D_t$ and $D_{1:t-1} = \{D_j\}_{j=1}^{t-1}$ arrive in sequence (Parisi et al., 2019). We define a population loss over the distribution $D_t$ by $E_{D_t}(\theta) = \mathbb{E}_{(x,y) \sim D_t}[L(f_\theta(x), y)]$, where $f_\theta(\cdot)$ is the model parameterized by $\theta$, and $L$ is a bounded loss function. The purpose of continual learning is to find a solution $\theta$ in a parameter space $\Theta$ that can minimize both $E_{D_t}(\theta)$ and $E_{D_{1:t-1}}(\theta)$ as much as possible with no access to old training samples. $\hat{E}_{D_{1:t-1}}(\theta_{1:t})$ is the robust empirical risk by the worst case of the neighborhood in parameter space aimed at finding a flat solution:

$$\hat{E}_{D_t}^r(\theta) = \max \hat{E}_{D_t}(\theta + \Delta), \frac{1}{2}\Delta^\top \Lambda_{k-1}\Delta \le r^2 \tag{1}$$

where $r$ is the radius around $\theta$ and $\Lambda_{k-1}$ is a hessian matrix. Recent studies suggest that flatter solutions are more robust to catastrophic forgetting (Cha et al., 2021; Deng et al., 2021; Jiang et al.,

2022). Based on the theoretical derivation of the work (Wang et al., 2022b), we can obtain the upper bound of the two losses. For any $\delta \in (0, 1)$ with probability at least $1 - \delta$, for every solution $\theta_{1:t}$ of the continually learned $1 : t$ tasks in parameter space $\Theta$, i.e., $\theta_{1:t} \in \Theta$:

$$E_{D_t}(\theta_{1:t}) < \hat{E}_{D_{1:t-1}}(\theta_{1:t}) + \frac{1}{2(t-1)} \sum_{j=1}^{t-1} \text{Div}(D_j, D_t) + \sqrt{\frac{d \ln(N_{1:t-1}/d) + \ln(1/\delta)}{N_{1:t-1}}} \quad (2)$$

$$E_{D_{1:t-1}}(\theta_{1:t}) < \hat{E}_{D_t}(\theta_{1:t}) + \frac{1}{2(t-1)} \sum_{j=1}^{t-1} \text{Div}(D_t, D_j) + \sqrt{\frac{d \ln(N_t/d) + \ln(1/\delta)}{N_t}} \quad (3)$$

$\text{Div}(D_i, D_j)$ represents the $H$-divergence between distribution $D_i$ and $D_j$, which quantifies the overall distribution differences between them. The third term is related to the dimensionality of the model's parameter space. Here, $d$ is the dimension of the parameter space $\Theta$, and $N_{1:t-1} = \sum_{k=1}^{t-1} N_k$ is the total number of training samples over all old tasks. Many CL methods have been proposed in recent years, which are separated into three types: replay-based methods, dynamic network-based methods, and optimization-based methods. The loss function for them can typically be defined as:

$$L(\theta) = L_t(\theta) + \lambda \hat{L}_{1:t-1}(\theta) \quad (4)$$

where $\hat{L}_{1:t-1}(\cdot)$ provides the constraint to achieve a proper trade-off between new and old tasks.

Replay-based methods (Luo et al., 2024; Rebuffi et al., 2017; Shin et al., 2017; Channappayya et al., 2024; Zhou et al., 2022a) facilitate continual learning by storing and replaying, or generating learned samples. $\hat{L}_{1:t-1}(\cdot)$ of them is $\sum_{k=1}^{t-1} L_k(\theta; \hat{D}_k)$, where $\hat{D}_k$ is an approximation of $D_k$ through replaying old training samples. Although these methods are effective, they lead to uncontrolled growth in storage and computational resource requirements and suffer from sample imbalance with tasks added. This imbalance can cause interference, as tasks with more replay samples affect learning new tasks and those with fewer samples.

Dynamic network-based methods (Bonato et al., 2024; Yoon et al., 2017; Wang et al., 2022a; Mallya & Lazebnik, 2018; Hu et al., 2023; Yan et al., 2021) primarily achieve continual learning by adding new parameters for new tasks to varying degrees while freezing old parameters. $\hat{L}_{1:t-1}(\cdot)$ of them is $\hat{L}_{1:t-1}(\theta = \bigcup_{k=1}^{t-1} \hat{\theta}_k)$. For every task, $\theta = \{\hat{\theta}_{old}, \hat{\theta}_{new}\}$, where $\hat{\theta}_{old}$ decides the extent to which frozen parameters from old tasks are reused varies across methods. In parameter isolation approaches (Yoon et al., 2017), $\hat{\theta}_{old}$ is zero, while in network expansion methods (Wang et al., 2022a), all frozen parameters are reused. When using a shared set of parameters across all tasks, the dimensionality $d$ is larger than when each task has its smaller set of parameters. These methods primarily aim to minimize the $\sqrt{\frac{d \ln(N_{1:t-1}/d) + \ln(1/\delta)}{N_{1:t-1}}}$ to reduce the upper bound of the loss function. While these methods effectively maintain the model's performance on new and old tasks, they do not enable backward transfer during learning, and networks grow uncontrollably with tasks added.

Optimization-based methods (Kao et al., 2021; Saha et al., 2021; Saha & Roy, 2023; Zeng et al., 2019; Lin et al., 2022; Kirkpatrick et al., 2017; Li & Hoiem, 2017; Yu et al., 2020) achieve continual learning by restricts parameter updates to directions which do not interfere strongly with previous tasks. $\hat{L}_{1:t-1}(\cdot)$ of them is $\hat{L}_{1:t-1}(\theta, \Lambda_{k-1})$. These methods are roughly equivalent to Eq. 1, which uses the Hessian matrix to constrain the updates of new tasks. $\Lambda_{k-1}$ is challenging to compute, it is often approximated by Fisher Information Matrix (FIM) (Liu et al., 2020; Spall, 2005):

$$F_k = E_{p(\hat{D}_k|\theta)} \left[ \nabla_\theta \log p(\hat{D}_k|\theta) \nabla_\theta \log p(\hat{D}_k|\theta)^\top \right] \Bigg|_{\theta=\mu_k} \approx \Lambda(D_k, \mu_k) \quad (5)$$

$F_k$ represents the Fisher Information Matrix, which measures the sensitivity of the parameter $\theta$ to the uncertainty during training (Kao et al., 2021). $\nabla_\theta \log p(x|\theta)$ is the gradient of the log-likelihood function concerning the parameter $\theta$. Different methods employ varying approaches to approximate the FIM (Zeng et al., 2019; Lin et al., 2022; Kirkpatrick et al., 2017; Li & Hoiem, 2017; Yu et al., 2020). While this method effectively avoids backward interference and promotes forward transfer, it hinders backward transfer. Additionally, when important parameters of the new and old tasks overlap, it greatly reduces the model's plasticity for new tasks, which is forward inference.

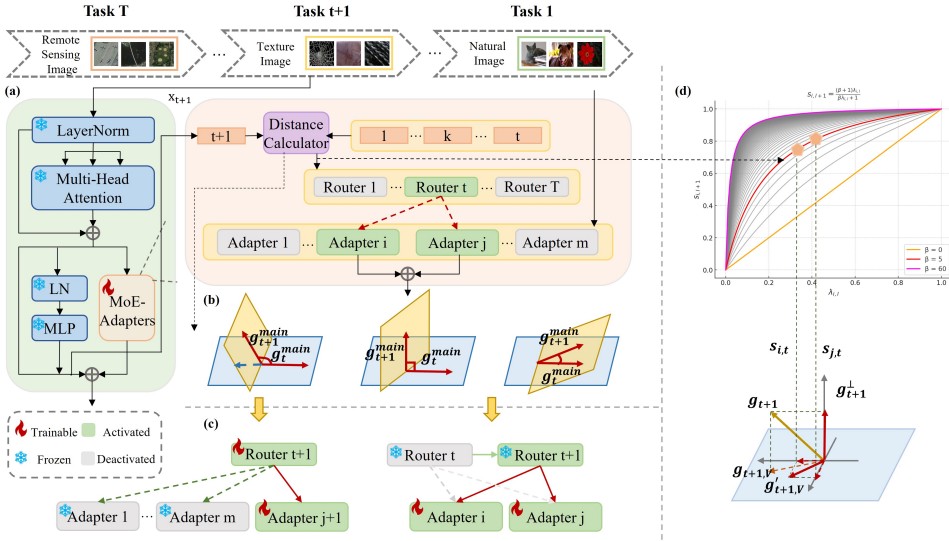

Figure 2: **Illustration of our method.** **(a):** The structure of the model and its operational state during training. **(b):** The conditions for transfer and interference occurs between tasks. **(c):** The different strategies the model employs to construct new parameter spaces for new tasks in response to either interference or transfer. **(d):** The principle of Dynamic Gradient Updates.

Compared to traditional approaches trained from scratch, PTM-based CL methods use a frozen pre-trained model as initialization and combine fine-tuning techniques and methods mentioned to adapt to new tasks (Zhou et al., 2024b; Liang & Li, 2024; Yu et al., 2024; Qiao et al., 2023). Some approaches learn a prompt pool to adaptively select instance-specific prompts for model updates (Wang et al., 2022d;c; Smith et al., 2023; Zhou et al., 2024a). Other representation-based methods leverage the generalization power of PTMs to construct classifiers (Zhou et al., 2024a) directly. However, few methods mentioned above can detect all types of transfer and interference, which prevents them from maximizing transfer and avoiding interference effectively.

## 3 THE CONDITIONS FOR TRANSFER AND INTERFERENCE

Eq. 1 measures the flatness of the loss surface around the solution, indicating the sensitivity of the loss function to parameter updates. The so-called flat direction refers to the direction in which the model is less sensitive, the corresponding element in the FIM is relatively small (Kao et al., 2021; Achille et al., 2019). Thus, the FIM is not only task-related but also dependent on specific model parameters. As seen above, when the model receives a new task, to minimize both $E_{D_t}(\theta_{1:t})$ and $E_{D_{1:t-1}}(\theta_{1:t})$, the new model should be optimized along the flatter directions:$\Delta = \arg\min_{\Delta} L_t(\theta) + \nabla_\theta L_t(\theta)^\top \Delta$ subject to $\frac{1}{2}\Delta^\top FIM_{t-1}\Delta \leq r^2$,, where $L_t(\theta + \Delta) \approx L_t(\theta) + \nabla_\theta L_t(\theta)^\top \Delta$ is a first-order approximation to the updated Laplace objective (Kao et al., 2021). Through derivation in Appendix A.1, we get update rules of $\theta$:

$$\theta \leftarrow \theta - \lambda FIM_{t-1}^{-1}\nabla_\theta L_t(\theta) \tag{6}$$

From Eq. 6, we can deduce that the optimization of new tasks is constrained along directions that are sensitive to prior tasks, while optimization in less sensitive directions is unrestricted. As shown in Fig. 2 (a), if the optimization direction of the new task opposes that of the prior task along sensitive parameters, updating in this new direction will degrade the prior task's performance, causing backward interference. On the other hand, constraining optimization in this direction may prevent reaching the optimal solution, leading to forward interference. When the optimization directions of the new and prior tasks share a common component in sensitive parameters, the alignment of their optimization directions allows for parameter reuse, promoting forward transfer. Furthermore, slight updates in this direction for the new task may improve the prior task's performance, resulting in

backward transfer. Therefore, we define transfer and interference in local model modules during continual learning as follows:

**Defination 1.** Let $R_{k,z}^{\theta}$ be the interaction between task $k$ and task $z$ on parameters $\theta$. $P_k$ are sensitive parameters in $FIM_k$, $P_{\cap} = P_k \cap P_z$. $g_{i,k}$ is gradient direction of task $k$ on parameters $p_i$. For any $\theta$ of model:

$$R_{k,z}^{\theta} = \begin{cases} Transfer, & \text{if } P_{\cap}! = 0, \exists p_i \in P_{\cap}, g_{i,k} \cdot g_{i,z} > 0 \\ Inference, & \text{if } P_{\cap}! = 0, \exists p_i \in P_{\cap}, g_{i,k} \cdot g_{i,z} < 0 \\ No\ relevance, & \text{if } P_{\cap} = 0 \text{ or } P_{\cap}! = 0, \forall p_i \in P_{\cap}, g_{i,t} \cdot g_{i,t-1} = 0 \end{cases} \quad (7)$$

## 4 METHODS

This paper deals with class-incremental learning, where tasks with disjoint data label spaces and task identities are only provided in training. We propose an adaptive continual learning approach based on fine-tuning adapters added to a pre-trained ViT model. This approach can actively detect transfer and interference and apply corresponding strategies. We first provide an overview of the fine-tuning adapter scheme used. Next, based on the derived conditions for transfer and interference, we introduce a task transfer distance metric. We then introduce how to use this metric to assess the transfer distance between new and old tasks and apply different continual learning strategies. They share the same adapter and apply dynamic gradient updates when tasks show high transfer or low relevance. Otherwise, they introduce new adapters and activate the frozen old adapters to assist new task learning in cases of high interference. More details about methods are in Appendix A.3.

### 4.1 SPACE EXPANSION WITH ADAPTERS

As shown in Fig. 2a, unlike previous methods that maintain an independent adapter for each task to support model expansion (Tan et al., 2024), we insert a set of adapters within each transformer block and employ a Mixture of Experts (MoE) mechanism (Masoudnia & Ebrahimpour, 2014; Du et al., 2022; Zhou et al., 2022b) for each task. We use LoRA (Gupta, 2021; Ding et al., 2023) as our adapter. As shown in Fig. 2 (a), a dedicated router is used for each task to select the appropriate adapters to activate. Since the task ID is not provided during inference, we learn a class center for each task during training. Then, during inference, we calculate the closest class center to the sample and use it to select the corresponding router. During the learning of task $t$, the activated adapters are fine-tuned for the new task, while the pre-trained weights $W$ and other adapters remain frozen. Expanding the parameter space through adapter combinations creates a flexible framework that supports various strategies for both transfer and inference, improving the model's adaptability.

### 4.2 TRANSFER DISTANCE EVALUATION

According to Definition 1, the occurrence of transfer or interference between tasks depends on whether the sensitive parameters of the new task overlap with those of previous tasks, as well as the direction of the gradient updates on these overlapping parameters. Using Eq. 6, we identify a task's sensitive parameters through the Fisher Information Matrix (FIM). However, calculating the FIM requires network activation based on task data, which can pose challenges if the network is not well-trained. In contrast, we leverage pre-trained models with strong feature extraction capabilities and high generalization during continual learning. Thus, we use the frozen pre-trained backbone as the FIM estimator. Since we fine-tune only a small number of parameters while keeping most frozen, the FIM derived from the pre-trained model is highly representative and can effectively guide adapter updates. As shown in Fig. 2 (a), we employ the frozen pre-trained backbone as a feature extractor and re-train a classifier for each block on the given task, which is typically efficient. After training, we compute the FIM for each block.

Since the full FIM is too large for transformer-based blocks, we focus only on the diagonal entries. To prevent noise when training with limited samples, instead of direct computation, we use a more robust estimator inspired by variational inference, as described in Achille et al. (2019). Assume we perturb the network weights $\hat{\theta}$ with Gaussian noise $\mathcal{N}(0, \Lambda)$, where $\Lambda$ is the precision matrix. Our goal is to find the optimal $\Lambda$ that minimizes the expected error while staying close to an isotropic prior $\mathcal{N}(\theta_w, \lambda^2 I)$. Specifically, we aim to find $\Lambda$ that minimizes: $L(\hat{\theta}; \Lambda) =$

$\mathbb{E}_{\theta \sim \mathcal{N}(\hat{\theta}, \Lambda)}\left[L(p_\theta, \hat{p}(y|x))\right] + \beta \, \text{KL}\left(\mathcal{N}(0, \Lambda) \| \mathcal{N}(0, \lambda^2 I)\right)$, where $\beta$ controls the weight of the prior, $KL$ is KL-divergence (Xie & Song, 2023; Vaitl et al., 2022). Approximating to the second order, the optimal value of $\Lambda$ satisfies $\frac{\beta}{2N}\Lambda = F + \frac{\beta\lambda^2}{2N}I$. Therefore, $\frac{\beta}{2N}\Lambda \sim F + o(1)$ can be considered as an estimator of the FIM. This estimator is easy to compute using Stochastic Gradient Variational Bayes (Achille et al., 2019).

Having accurately estimated the diagonal elements of the FIM, we now have the sensitivity of each parameter in the model. Next, we normalize the gradients $G = \{g_1, ...g_d\}$ of the samples obtained from the frozen modules to capture the gradient update direction $\hat{G} = \{\hat{g_1}, ...\hat{g_d}\}$ for parameters and task. By multiplying the parameter sensitivities by their corresponding gradient update directions, we obtain a task embedding that integrates both parameter sensitivity and gradient direction:

$$Emd_{\theta,i} = F_{\theta,i} * \hat{G_{\theta,i}} = \{\sigma_{\theta,i,1}\hat{g_1}, ...\sigma_{\theta,i,d}\hat{g_d}\} = \{emd_1, ...emd_d\} \quad (8)$$

where $Emd_{\theta,i}$ is the task embedding of task $i$ on weight $\theta$. we can compute transfer distance $TD_{i,j}$ of task $i$ and task $j$ by $TD_{\theta,i,j} = \sum_{k=1}^{d} emd_{\theta,i,k} \cdot emd_{\theta,j,k}$ We can see that when both tasks show high sensitivity to the same parameter and the product of their gradient update directions is positive, the transfer distance increases, indicating a greater degree of transfer between the tasks. Conversely, if the sensitivity rankings for the same parameter differ between the tasks, or if the product of their gradient update directions is negative, the transfer distance decreases, leading to greater interference between the tasks.

### 4.3 STRATEGIES BASED ON ACTIVE DETECTION OF TRANSFER AND INTERFERENCE

#### 4.3.1 MORE TRANSFER BETWEEN TASKS

As shown in Fig 2 (a), we compute the transfer distance metric between new and old tasks, and determine whether two tasks are similar or non-interfering. As Fig. 2 (c) shows, we make the new task share the same adapters and pathways with the old task with the highest transfer. We propose a dynamic gradient adjustment method based on transfer distance, which allows controlled updates, enhances forward knowledge transfer, and improves model generalization. It consists of three key components: extracting and updating the principal directions of prior tasks, calculating and updating the importance of these directions, and dynamically adjusting gradients for the new task.

**The extraction and update of principal directions.** When an adapter is activated for the first time, it is crucial to capture and record the key optimization directions of the current tasks with their importance for future updating. Therefore, we obtain the activations $A_t$ of adapters and perform SVD on them $A_t = U_t \Sigma_t V_t^T$, where $U_t$ and $V_t$ are orthonormal matrices, and $\Sigma_t$ has sorted singular values $(\sigma_{i,t})$ along its diagonal. According to Principal Component Analysis (PCA) (Abdi & Williams, 2010), we sort the singular values in descending order and select the top $z_t$ left singular vectors $U_t^{k_t}$ corresponding to the largest singular values, ensuring that it satisfies $\|A_t U_t^{k_t}\|_F^2 \geq \alpha \|A_t\|_F^2$, $\|.\|_F^2$ is the Frobenius norm (Cortinovis & Kressner, 2020; Xi, 2021) of the matrix. The threshold hyperparameter, $\alpha \in (0, 1)$ controls the value of $k_t$ selected. We store these bases in $V = [v_{1,t}, v_{2,1}, \ldots, v_{k_t,t}]$ as important directions for current task. After the end of task $t + 1$, we update $V$ by adding the important gradient space for this task. Since we utilize transfer distance evaluation, there may be overlapping feature vectors between task $t$ and task $t+1$. Thus, we eliminate redundant feature vectors from the task $t + 1$ and retain the new ones to add to the feature basis set. We first project the task $t + 1$ activations $A_{t+1}$ onto the complementary space represented by $V$:

$$A'_{t+1} = A_{t+1} - (VV^\top)A_{t+1} = A_{t+1} - A_{t+1,V} \quad (9)$$

Then SVD is performed on $A'_{t+1} = U'_{t+1}\Sigma'_{t+1}V'_{t+1}$ and new $k_{t+1}$ bases are chosen for minimum $k_{t+1}$ satisfying the criteria: $\|A_t U_t^{k_t}\|_F^2 + \|A_t U_t^{k_t}\|_F^2 \geq \alpha \|A_t\|_F^2$. Gradient space in V is updated (after $t + 1$ update) by adding these new bases to it.

**The computation and update of the importance of the main direction** Through transfer distance, we observe that both tasks show co-directional updates in these overlapping directions. Therefore, we can appropriately relax the constraints on gradient updates in the shared directions. The degree of relaxation depends on two key factors: (1) the sensitivity of the previous task in the overlapping

directions, and (2) the magnitude of the new task's updates in those directions. The singular values from matrix decomposition indicate the importance of the corresponding singular vectors. Therefore, we determine the sensitivity of each optimization direction, represented by the basis vectors, as $\lambda_i = \frac{\sigma_{i,t}}{max(\sigma_t)}$, based on their singular values. After learning the new task $t+1$, we propose a method for finding and updating the importance without using data from old tasks. As discussed previously, we obtain new feature vectors by performing Singular Value Decomposition (SVD) (Abdi, 2007) on the projection of $A_{t+1}$ onto $(1 - VV^\top)$. The corresponding singular values represent the importance of these feature vectors. However, we cannot directly derive the sensitivity of $V$ for task $t+1$ from this. Given that the task $t+1$ may contain redundant feature vectors that are linear combinations of $V$, we first compute the coordinates of the redundant vectors in $U_{t+1,V}$: $C = V^T U_{t+1,V}$. Then, by multiplying these coordinates with the corresponding singular values of the redundant vectors, we obtain the task's sensitivity to the basis: $\sigma'_V = \sqrt{(C \odot C)(\sigma_{t+1,V})^2}$, here $\odot$ denotes element-wise multiplication (Lee et al., 2021), $(.)^2$ and $\sqrt{(.)}$ denote element-wise square and square root operations respectively. Then we get new single values for task $t+1$: $\sigma_{t+1} = \begin{bmatrix} \sigma'_V \\ \sigma_{t+1} \end{bmatrix}$. Therefore, we can use $\sigma_{t+1}$ to obtain the basis importance vector $\lambda_{t+1} = \frac{\sigma_{i,t+1}}{max([\sigma'_V, \sigma'_{t+1}])}$ for the given $i^{th}$ basis.

Finally, we update the importance of old $k$ bases by: $\lambda_i = \begin{cases} \lambda_i, & \text{if } \lambda_i \geq \lambda_{i,t+1} \\ \lambda_{i,t+1}, & \text{otherwise} \end{cases}$ We then add the importance of new bases in $\lambda$ as $[\lambda', \lambda_{t+1,V}]$.

**Dynamic gradient updates.** we propose Dynamic Gradient Updates (DGU) to adjust the gradient updates along the prior task's basis, using the importance parameters of the prior task and the transfer distance between the new and prior tasks. We then compute scaling factor for $i^{th}$ basis, by following:

$$s_{i,t+1} = \frac{(\beta + 1)\lambda_{i,t}}{\beta\lambda_{i,t} + 1} \tag{10}$$

where $\beta$ is a non-negative scale coefficient hyperparameter. The value of $s_{i,t+1}$ will range from 0 to 1 as we are concerned with the non-negative singular values. Eq. 10 ensures that a maximum importance of 1 is assigned to the basis with the highest singular value and other bases are given importance $(< 1)$ relative to this maximum. In our formulation, $\lambda_{i,t} = 1$ means no gradient step is allowed along the corresponding basis direction for the new tasks, whereas along other basis gradients are scaled by the factor of $(1 - \lambda_{i,t})$. We allow a scaled gradient update along those bases (Figure 1(d)) enabling higher plasticity for new tasks, while importance-based scaling ensures adequate stability of past tasks. As shown in Fig. 2 (d), scaled gradient updates along those bases, which increases plasticity for new tasks, while importance-based scaling maintains stability for previous tasks. As shown in Fig. 2 (d), as the $\beta$ increases, all scaling factors approach 1, mimicking the behavior of traditional projection-based methods that block optimization of the new task on a prior basis. Given this characteristic, we implement a phased approach to adjust the parameter based on the transfer distance between tasks:

$$\beta = \begin{cases} \beta > \beta_{th}, & \text{if } dis \geq th \\ \beta < \beta_{th}, & \text{otherwise} \end{cases} \tag{11}$$

This parameter control allows us to regulate the extent of optimization for new tasks with highly overlapping optimization directions, ensuring a balance between tasks, improving generalization, and preventing any single model from dominating in specific directions. We learn the $t+1$ task sequentially using only its dataset, $D_{t+1}$. Let $L_{t+1}$ represent the loss for the $t+1$ task. To prevent catastrophic forgetting and enable new learning, we apply a scaled gradient projection to the new gradients, $\nabla W_{t+1} L_{t+1}$, as follows: $\nabla W_{t+1} L_{t+1} = \nabla W_{t+1} L_{t+1} - (V \Sigma V^\top)(\nabla W_{t+1} L_{t+1})$, As Fig. 2 (d) shows, it ensures the gradient components along orthogonal directions to $V$ will not be changed, while the importance scaled gradient components will be scaled by $(1 - \lambda_1)$.

### 4.3.2 MORE INTERFERENCE BETWEEN TASKS

When the transfer distance is too small, it means their feature spaces overlap, but the gradient update directions in these overlapping areas are opposite. In this situation, updating the new task along these shared gradient directions would severely degrade the performance of previous tasks, making parameter sharing unsuitable. Therefore, we introduce a new sub-parameter space adapter for the

new task to act as its primary adapter. Meanwhile, all old adapters are frozen, and the routing mechanism is trained to select a few old branches with lower importance to participate in learning the new task.

# 5 EXPERIMENTS

In this section, we first compared our method to SOTA methods and typical CL methods on seven benchmark. We then conducted experiments under different settings to validate the robustness of our algorithm. Additionally, an ablation study was performed to assess the effectiveness of each component of our method. Finally, we analyzed transfer phenomena and shared weights in experiments, demonstrating that our proposed method can maximize transfer and avoid interference.

## 5.1 IMPLEMENTATION DETAILS

**Dataset and Settings.** We follow (Zhou et al., 2024a) to evaluate the performance on three datasets with the overlap between pre-trained datasets and four datasets with large domain gap with it, which are CIFAR100 (Krizhevsky et al., 2009), CUB200 (Wah et al., 2011), ImageNet-R (Hendrycks et al., 2021a), ImageNet-A (Hendrycks et al., 2021b), ObjectNet (Barbu et al., 2019), Omnibenchmark (Zhang et al., 2022) and VTAB (Zhai et al., 2019). We use 'B-m Inc-n' to represent the configuration where m classes are in the base stage and n classes in each incremental stage. **Comparison methods.** We compare our method to state-of-the-art PTM-based CL methods, including L2P (Wang et al., 2022d), DualPrompt (Wang et al., 2022c), CODA-Prompt (Smith et al., 2023), SimpleCIL (Zhou et al., 2024a) and ADAM (Zhou et al., 2024a). Additionally, we evaluate it against typical continual learning methods adapted with PTM, such as LwF (Li & Hoiem, 2017), SDC (Yu et al., 2020), iCaRL (Rebuffi et al., 2017), DER (Yan et al., 2021), FOSTER (Wang et al., 2022a) and MEMO (Zhou et al., 2022a). We also report the baseline methods: sequential PTM finetuning (Finetune) and PTM finetuning with adapters (Finetune Adapter). All methods are implemented using the same PTM. **Evaluation metric.** Following (Zhou et al., 2024a), we use $A_b$ to denote the model's accuracy after the $b$-th stage. Specifically, we measure $A_B$ (accuracy after the final stage) and $\bar{A} = \frac{1}{B} \sum_{b=1}^{B} A_b$ (average accuracy across all stages). More details in Appendix A.4

## 5.2 COMPARISON WITH OTHER METHODS

Table 1: Average and last performance comparison on seven datasets with ViT-B/16-IN21K as the backbone. 'IN-R/A' stands for 'ImageNet-R/A,' 'ObjNet' stands for 'ObjectNet,' and 'OmniBench' stands for 'OmniBenchmark.' '*' means we get outcomes in published work.

| Method | CIFAR B0 Inc5 | | CUB B0 Inc5 | | IN-R B0 Inc5 | | IN-A B0 Inc20 | | ObjNet B0 Inc5 | | OmniBench B0 Inc30 | | VTAB B0 Inc10 | |
|---|---|---|---|---|---|---|---|---|---|---|---|---|---|---|
| | $\bar{A}$ | $A_B$ | $\bar{A}$ | $A_B$ | $\bar{A}$ | $A_B$ | $\bar{A}$ | $A_B$ | $\bar{A}$ | $A_B$ | $\bar{A}$ | $A_B$ | $\bar{A}$ | $A_B$ |
| Finetune | 38.90 | 20.17 | 26.08 | 13.96 | 21.61 | 10.79 | 24.28 | 14.51 | 19.14 | 8.7 | 23.61 | 10.57 | 34.95 | 21.25 |
| Finetune Adapter* | 60.51 | 49.32 | 46.12 | 52.99 | 47.59 | 40.28 | 47.50 | 41.10 | 50.22 | 35.95 | 62.32 | 50.53 | 48.91 | 45.12 |
| LwF* | 46.29 | 41.07 | 48.97 | 32.03 | 39.93 | 26.47 | 37.75 | 26.84 | 33.01 | 20.65 | 47.14 | 33.95 | 40.48 | 27.54 |
| SDC* | 68.21 | 63.05 | 70.62 | 66.37 | 52.17 | 49.20 | 29.11 | 26.63 | 39.04 | 29.06 | 60.94 | 50.28 | 45.06 | 22.50 |
| L2P* | 85.94 | 79.93 | 67.05 | 56.25 | 66.53 | 59.22 | 49.39 | 41.47 | 63.78 | 52.19 | 73.36 | 64.69 | 77.11 | 70.10 |
| DualPrompt* | 87.87 | 81.15 | 71.47 | 66.54 | 63.31 | 55.22 | 53.71 | 41.67 | 59.27 | 49.33 | 73.92 | 65.52 | 83.36 | 81.23 |
| CODA-Prompt* | 89.11 | 81.96 | 84.00 | 73.37 | 64.42 | 55.08 | 53.54 | 42.73 | 66.07 | 53.29 | 77.03 | 68.09 | 83.90 | 83.02 |
| SimpleCIL* | 87.57 | 81.26 | 92.20 | 86.73 | 62.58 | 54.55 | 59.77 | 48.91 | 65.45 | 53.59 | 79.34 | 73.15 | 85.99 | 84.38 |
| ADAM + Finetune* | 87.67 | 81.27 | 91.82 | 86.39 | 70.51 | 62.42 | 61.01 | 49.57 | 61.41 | 48.34 | 73.02 | 65.03 | 87.47 | 80.44 |
| ADAM + VPT-S* | 90.43 | 84.57 | 92.02 | 86.51 | 66.63 | 58.32 | 58.39 | 47.20 | 64.54 | 52.53 | 79.63 | 73.68 | 87.15 | 85.36 |
| ADAM + VPT-D* | 88.46 | 82.17 | 91.02 | 84.99 | 68.79 | 60.48 | 58.48 | 48.52 | 67.83 | 54.65 | 81.05 | 74.47 | 86.59 | 83.06 |
| ADAM + SSF* | 87.78 | 81.98 | 91.72 | 86.13 | 68.94 | 60.60 | 61.30 | 50.03 | 69.15 | 56.64 | 80.53 | 74.00 | 85.66 | 81.92 |
| ADAM + Adapter* | 90.65 | 85.15 | 92.21 | 86.73 | 72.35 | 64.33 | 60.47 | 49.37 | 67.18 | 55.24 | 80.75 | 74.37 | 85.95 | 84.35 |
| EASE | 91.51 | 85.80 | **92.23** | 86.81 | **78.31** | 70.58 | 65.34 | 55.04 | **70.84** | 57.86 | 81.11 | 74.85 | 93.61 | 93.55 |
| **ours** | **93.34** | **89.20** | 92.06 | **90.37** | 76.20 | **73.28** | **67.61** | **59.71** | 69.45 | **59.13** | **83.26** | **75.46** | **95.46** | **94.11** |

**Benchmark comparison.** We compare our proposed method with other state-of-the-art approaches across seven benchmark datasets using different backbone weights. Tab. 1 shows the results using ViT-B/16-IN21K, where our method achieves the best performance on all seven benchmarks, significantly surpassing current SOTA methods. Figure 1 illustrates the incremental performance trends using ViT-B/16-IN1K. As indicated in each image, our method outperforms the runner-up by $0.7\% to 2\%$ on ImageNet-R/A, ObjectNet, OmniBenchmark, and VTAB. **Comparison with typical CL methods.** We also compare our method to typical CL approaches using the same pre-trained

model, as shown in Tab. 1. Unlike these typical CL methods, which require saving exemplars to retain previous knowledge, our method does not. Following the setup from (Rebuffi et al., 2017), we find that, surprisingly, our method remains competitive even against exemplar-based approaches. **Parameters efficient.** We examine the number of parameters used by different methods and present the parameter-performance comparison on ImageNet-R B100 Inc50 in Fig. 1 (a). As shown in Fig. 1 (a), our method uses a similar number of parameters as other prompt-based methods and EASE, yet achieves the highest performance among all competitors. This highlights that our proposed method strikes a better balance between parameter efficiency and accuracy compared to other algorithms.

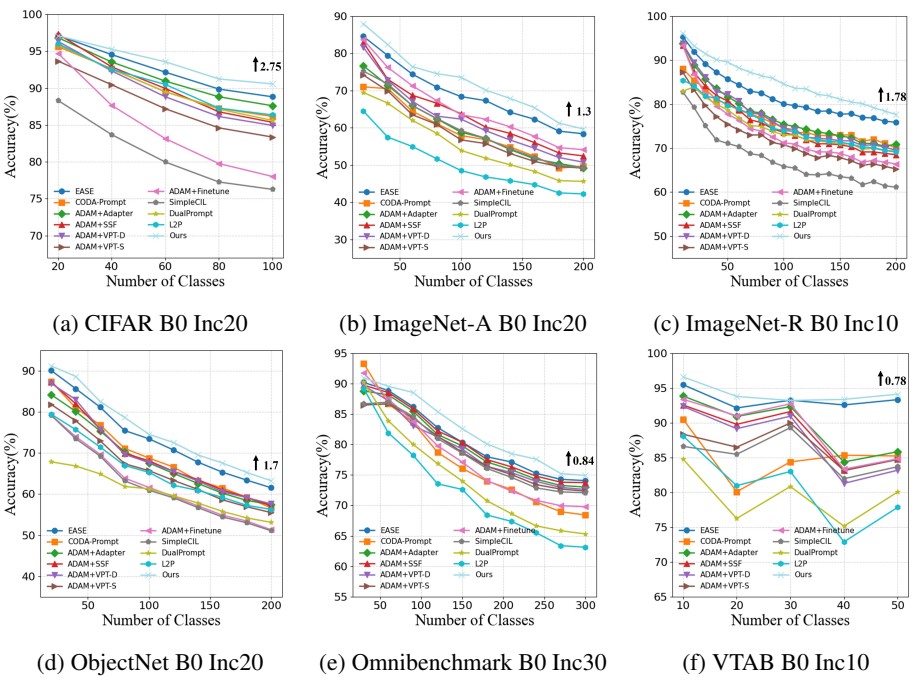

Figure 3: Performance comparison across different benchmarks.

## 5.3 ABLATION STUDY

We conduct an ablation study to assess the effectiveness of each component in our proposed method. We report the incremental performance and parameters efficiency of variations on the VTAB, which has significant category differences, as shown in Fig. 1 (b). **Effectiveness of MoE.** We present the performance of "MoE" which applies the MOE mechanism across ten groups of adapters. Even without additional continual learning techniques to prevent catastrophic forgetting on shared parameters, the MoE mechanism alone learns task-specific paths and adapters, helping retain part of the model's performance. Notably, as Fig. 1 (b) and Tab. 3 shows, the performance of "MoE" surpasses certain continual learning methods based on pre-trained models, further demonstrating its effectiveness. **Effectiveness of DGU and independent adapters.** We further remove the transfer distance evaluation and evaluate two variations: one uses dynamic gradient updates (DGU) to optimize two shared adapters, and the other maintains two independent adapters for each task. These variations are labeled "DGU" and "Independent" respectively. Fig. 1 (b) shows that "DGU" significantly outperforms the "MoE" method, indicating that DGU alone has strong continual learning capabilities. **Effectiveness of actively detecting transfer and interference.** As Fig. 1 (b) shows, our method, which combines transfer distance evaluation with DGU and independent adapters, achieves accuracy levels notably higher than DGU alone and is comparable to "Independent". However, our approach requires far fewer trainable parameters compared to 'independent'. This demonstrates that the introduction of active transfer and interference detection allows the model to apply more effective continual learning strategies, maximizing beneficial transfer, and reducing interference.

Table 2: Comparison to typical CL methods. All methods are based on the same pre-trained model.

| Method | Exemplars | ImageNet-R B0 Inc20 | | CIFAR B0 Inc10 | |
|---|---|---|---|---|---|
| | | $\bar{A}$ | $A_B$ | $\bar{A}$ | $A_B$ |
| iCaRL* | 20 / class | 72.42 | 60.67 | 82.46 | 73.87 |
| DER* | 20 / class | 80.48 | 74.52 | 86.04 | 77.93 |
| FOSTER* | 20 / class | 81.34 | 74.88 | 89.87 | 84.91 |
| MEMO* | 20 / class | 74.80 | 66.62 | 84.08 | 75.79 |
| **Ours** | 0 | **84.23** | **79.12** | **95.42** | **89.65** |

Table 3: The number of adapters in different blocks of the model our proposed method learned during training. We use 2 adapters for a task.

| Settings | Number of Adapters | | | | | | | | | | | |
|---|---|---|---|---|---|---|---|---|---|---|---|---|
| | 1 | 2 | 3 | 4 | 5 | 6 | 7 | 8 | 9 | 10 | 11 | 12 |
| CIFAR B0 Inc5 | 2 | 2 | 2 | 2 | 2 | 2 | 2 | 2 | 2 | 2 | 2 | 2 |
| CUB B0 Inc5 | 2 | 2 | 2 | 2 | 2 | 19 | 16 | 18 | 20 | 20 | 20 | 2 |
| IN-R B0 Inc5 | 3 | 3 | 3 | 3 | 3 | 3 | 3 | 3 | 3 | 4 | 5 | 3 |
| IN-A B0 Inc20 | 2 | 2 | 2 | 2 | 2 | 4 | 2 | 2 | 2 | 5 | 5 | 2 |
| ObjNet B0 Inc5 | 5 | 6 | 5 | 5 | 5 | 6 | 15 | 12 | 16 | 15 | 18 | 5 |
| OmniBench B0 Inc30 | 9 | 9 | 10 | 10 | 10 | 6 | 6 | 2 | 2 | 2 | 2 | 2 |
| VTAB B0 Inc10 4 | 4 | 4 | 4 | 4 | 4 | 4 | 4 | 4 | 4 | 4 | 4 | 4 |

## 5.4 TRANSFER AND INFERENCE ANALYSIS

As shown in the Fig. 1 (c) and Tab. 3, we provide the number of adapters used in different blocks of the model under various settings, along with the individual performance of our method and its variants on all tasks in the VTAB dataset. Tab. 3 shows that, in our method, two tasks share parameters at each layer on the VTAB. By tracking the training process, we found that stages 1 and 4, both involving satellite remote sensing images, shared parameters, as did stages 3 and 5, which both focused on natural images. This demonstrates that our algorithm accurately captures task-specific transfer and interference. Fig. 1 (c) highlights that our method significantly outperforms other variants in stages 1 and 3, indicating that our approach not only detects transfer but also maximizes backward knowledge transfer through effective CL strategies. The table also reveals a clear layering pattern in parameter sharing across different blocks. In some cases, earlier blocks have high-level sharing, suggesting similar low-level features, while later blocks show more sharing, reflecting similar high-level features. For VTAB, consistent parameter sharing across all layers suggests a strong domain-specific pattern within the dataset. In summary, our method enables the model to actively detect transfer and interference across different modules and tasks, and adapt continual learning strategies accordingly, maximizing knowledge transfer while minimizing interference.

## 6 CONCLUSION

This paper addresses the issue that most continual learning methods do not actively detect transfer or interference during learning, which prevents them from maximizing transfer or minimizing interference. We conduct a theoretical analysis to identify the conditions under which transfer and interference occur in continual learning. Based on this, we propose a method to measure task transfer and interference using pre-trained models. Furthermore, we introduce different strategies to handle transfer and interference. Our baseline experiments demonstrate that our algorithm can actively detect these phenomena during continual learning and apply appropriate strategies to maximize transfer and avoid interference.

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

# A APPENDIX

## A.1 THEORETICAL PROOF

### DERIVATION PROCESS

Given the following equation:

$$\Delta = \arg\min_{\Delta} L_t(0) + \nabla_\theta L_t(0)^T \Delta \quad \text{subject to} \quad \frac{1}{2}\Delta^T F_{IM_{t-1}}\Delta \le r^2$$

### 1. LAGRANGE MULTIPLIER METHOD

We introduce the Lagrange multiplier $\lambda$ to handle the constraint:

$$\mathcal{L}(\Delta,\lambda) = L_t(0) + \nabla_\theta L_t(0)^T \Delta + \lambda\left(\frac{1}{2}\Delta^T F_{IM_{t-1}}\Delta - r^2\right)$$

Taking the derivative with respect to $\Delta$ and setting it equal to 0:

$$\frac{\partial \mathcal{L}}{\partial \Delta} = \nabla_\theta L_t(0) + \lambda F_{IM_{t-1}}\Delta = 0$$

Solving for $\Delta$:

$$\Delta = -\frac{1}{\lambda} F_{IM_{t-1}}^{-1} \nabla_\theta L_t(0)$$

## 2. SOLVING FOR $\lambda$

Using the constraint $\frac{1}{2}\Delta^T F_{IM_{t-1}}\Delta \leq r^2$, substitute $\Delta$:

$$\frac{1}{2}\left(-\frac{1}{\lambda}\nabla_\theta L_t(0)\right)^T F_{IM_{t-1}}\left(-\frac{1}{\lambda}F_{IM_{t-1}}^{-1}\nabla_\theta L_t(0)\right) \leq r^2$$

Simplifying:

$$\frac{1}{2\lambda^2}\nabla_\theta L_t(0)^T F_{IM_{t-1}}^{-1}\nabla_\theta L_t(0) \leq r^2$$

Solving for $\lambda$:

$$\lambda^2 = \frac{1}{2r^2}\nabla_\theta L_t(0)^T F_{IM_{t-1}}^{-1}\nabla_\theta L_t(0)$$

Thus:

$$\lambda = \sqrt{\frac{1}{2r^2}\nabla_\theta L_t(0)^T F_{IM_{t-1}}^{-1}\nabla_\theta L_t(0)}$$

## 3. FINAL UPDATE RULE

Substituting $\lambda$ back into the expression for $\Delta$, we get the parameter update rule:

$$\Delta = -\frac{r}{\sqrt{\frac{1}{2}\nabla_\theta L_t(0)^T F_{IM_{t-1}}^{-1}\nabla_\theta L_t(0)}}F_{IM_{t-1}}^{-1}\nabla_\theta L_t(0)$$

it is often approximated by Fisher Information Matrix (FIM) (Liu et al., 2020; Spall, 2005):

$$F_k = E_{p(\hat{D}_k|\theta)}\left[\nabla_\theta \log p(\hat{D}_k|\theta)\nabla_\theta \log p(\hat{D}_k|\theta)^\top\right]\Bigg|_{\theta=\mu_k} \approx \Lambda(D_k, \mu_k) \tag{12}$$

$F_k$ represents the Fisher Information Matrix, which measures the sensitivity of the parameter $\theta$ to the uncertainty during training (Kao et al., 2021). $\nabla_\theta \log p(x|\theta)$ is the gradient of the log-likelihood function concerning the parameter $\theta$.

the work by (Wang et al., 2022b) demonstrates that this method leads to a tighter upper bound on the generalization gap than independent adapters through $\sqrt{\frac{d\ln(N_t/d)+\ln(1/\delta)}{N_t}}$. See more details in Appendix A.

$$\max_{i\in[1,K]}\sqrt{\frac{d_i\ln(N_{1:t-1}/d_i)+\ln(2K/\delta)}{N_{1:t-1}}} + \sqrt{\frac{d\ln(N_{1:t-1}/d)+\ln(1/\delta)}{N_{1:t-1}}}, \tag{13}$$

$$\max_{i\in[1,K]}\sqrt{\frac{d_i\ln(N_t/d_i)+\ln(2K/\delta)}{N_t}} + \sqrt{\frac{d\ln(N_t/d)+\ln(1/\delta)}{N_t}}. \tag{14}$$

Comparing Eq. 3 and Eq. 14, we conclude that cooperating $k$ adapters facilitates a smaller generalization gap over the new and old tasks.

### A.2 LIMITATIONS OF OTHER METHODS IN HANDLING TRANSFER AND INTERFERENCE

L2P (Wang et al., 2022d)applies visual prompt tuning to continual learning by learning a prompt pool to select instance-specific prompts. DualPrompt (Wang et al., 2022c) introduces two types of prompts, namely, general and expert prompts. CODA-Prompt (Smith et al., 2023) further improves the prompt selection process by incorporating an attention mechanism. SimpleCIL (Zhou et al., 2024a) freezes the pre-trained weights and extracts the center of each class by averaging the embeddings within the same class, resulting in the most representative pattern of that class. ADAM (Zhou et al., 2024a) further advances this approach by comparing the performance of the prototype-based classifier with that of a fully fine-tuned model on new classes.

### A.3 DETAILS OF METHOD

**LoRA** (Gupta, 2021; Ding et al., 2023) includes a dimensionality reduction matrix $F_{down} \in \mathbb{R}^{l \times d}$ and a dimensionality increasing matrix $F_{up} \in \mathbb{R}^{d \times l}$: $o = Wx + \sum_{j=1}^{k} F_{up}^j F_{down}^j x$. where $x$ denotes inputs of the block, $o$ denotes outputs, $W$ is the frozen weight of pre-trained model and $k$ is the number of activated adapters.

**Effectiveness.** Based on Eq. 3, we analyze the theoretical effectiveness of our algorithm. First, for $\hat{E}_{D_{1:t-1}}(\theta_{1:t})$, our algorithm shares a set of parameters among tasks that fall within the same flat optimization region and applies a suitable flat direction search method, thereby tightening the upper bound of this term. For the second term, $\frac{1}{2(t-1)} \sum_{j=1}^{t-1} \text{Div}(D_j, D_t)$, since the FIM closely aligns with task similarities, reducing the divergence between them. Finally, the MoE mechanism also reduces the third term. In conclusion, our algorithm effectively tightens the upper bound of the loss function across all three aspects, enabling strong continual learning performance. The work by (Wang et al., 2022b) demonstrates that MOE-adapters lead to a tighter upper bound on the generalization gap than independent adapters through $\sqrt{\frac{d \ln(N_t/d) + \ln(1/\delta)}{N_t}}$.

**Estimate optimization directions.** Projection matrix-based methods estimate these optimization directions by using the principal components of network activations. Therefore, we first train the adapter using the current task's dataset $\mathcal{D}_t$. After training, we sample $S_t = [x_{1,t}, x_{2,t}, \dots, x_{n,t}]$ from the task dataset $\mathcal{D}_t$, obtain the activations $A_t = f_{adapter}(R_t)$ passing through the adapter, and perform SVD on the activations $A_t = U_t \Sigma_t V_t^T$, where $U_t$ and $V_t$ are orthonormal matrices, and $\Sigma_t$ has sorted singular values $(\sigma_{i,t})$ along its diagonal. We then extract the principal components via low-rank approximation based on $U_t$ and $\Sigma_t$. According to the proof from Principal Component Analysis (PCA) (Abdi & Williams, 2010), the larger the singular value, the more its corresponding left singular vector represents the primary information contained in the data. Therefore, we sort the singular values in descending order and select the top $z_t$ left singular vectors $U_t^{k_t}$ corresponding to the largest singular values for matrix dimensionality reduction, ensuring that it satisfies $\|A_t U_t^{k_t}\|_F^2 \geq \alpha \|A_t\|_F^2$, $\|.\|_F^2$ is the Frobenius norm (Cortinovis & Kressner, 2020; Xi, 2021) of the matrix. The threshold hyperparameter, $\alpha \in (0, 1)$ controls the value of $k_t$ selected. Saha et al. (Saha et al., 2021) showed that these bases equivalently span the most important gradient space. We store these bases in $V = [v_{1,t}, v_{2,1}, \dots, v_{k_t,t}]$ as important directions for current task. After the end of task $t + 1$, we update $V$ by adding the important gradient space for this task.

### A.4 EXPERIMENTS DETAILS

**Setting details.** VTAB contains 50 classes, CIFAR100 has 100 classes, CUB, ImageNet-R, ImageNet-A, and ObjectNet each have 200 classes, and OmniBenchmark includes 300 classes. To ensure a fair comparison, we use the same training and testing sets as in (Zhou et al., 2024a) for all methods. Following (Zhou et al., 2024a), we use two pre-trained models: ViT-B/16-IN21K and ViT-B/16-IN1K. Both are pre-trained on ImageNet21K, but the latter is further fine-tuned on ImageNet1K.

In our experimental setup, we assign two adapters for each task. For tasks requiring a new adapter, we currently set the number of frozen old branches to be reused to 1.

**Comparison in different settings.** In addition to the B0 settings shown in Tab. 1 and Fig. 3, we also conduct experiments with different base class configurations. As illustrated in Fig. 4 (a) and (b), our proposed method continues to perform competitively across various data split settings.

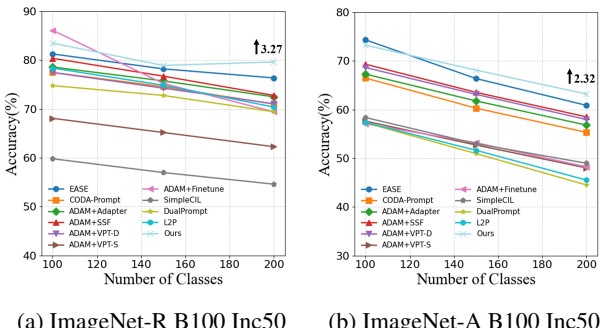

(a) ImageNet-R B100 Inc50      (b) ImageNet-A B100 Inc50

Figure 4: **Experiments results**. **(a):** Performance comparison with vase base stage on ImageNet-R B100 Inc50. **(b):** Performance comparison with vase base stage on ImageNet-A B100 Inc50.

