# OpenReview forum: "Adaptive Continual Learning Through Proactive Detection of Transfer and Interference"
_ICLR.cc/2025/Conference — Submitted to ICLR 2025_

### Official Review · Reviewer_bQ8S · 2024-10-31

**Soundness:** 2
**Presentation:** 1
**Contribution:** 2
**Rating:** 3
**Confidence:** 5

**Summary:**

This paper considers how to mitigate interference and forgetting between tasks in continual learning through the transfer of knowledge between tasks. The proposed method combines LoRA (Low-Rank Adaptation) and MoE (Mixture of Experts) to fine-tune pre-trained models.

**Strengths:**

The approach of performing continual learning through the transfer of knowledge between tasks is intuitive.

**Weaknesses:**

1. The writing in this paper is poor, making it difficult to follow.

2. The section "The extraction and update of principal directions" in Section 3.3.1 is essentially the GPM (ICLR 2021) [1] algorithm and should not be considered a contribution of this work. It should not be described in such detail.

3. It is recommended that the author include an overview section at the beginning of Section 3 to introduce the whole process of the proposed method, and at the end of Section 3, present Algorithm to summarize the whole process of the method.

4. There is a lack of a problem formulation section that explains what problem you are solving and what settings you are considering. Specifically, existing continual learning settings include task incremental learning, class incremental learning, domain incremental learning, and so on. This paper does not even mention what continual learning setting the authors are considering.

5. The experimental setup is not clearly described, such as learning rate, batch size, epoch, etc., are not provided. The hyperparameter settings for the method are also not clearly explained.

6. The writing in this article is difficult to follow. If the author can improve the writing and satisfactorily address the issues I am concerned about, I would consider raising the score.

[1] Saha G, Garg I, Roy K. Gradient projection memory for continual learning[J]. arXiv preprint arXiv:2103.09762, 2021.

**Questions:**

The authors only introduced the use of the MoE (Mixture of Experts) structure in Section 3.1, and from Figure 2, it can be seen that the proposed method maintains a task-specific router for each task. Since the router is task-specific, I would like to ask if the proposed method is addressing the class incremental learning problem. If so, since class incremental learning requires task labels are unavailable during the inference phase, how do the proposed method selects the task routers to use during inference for a given test sample?
If this is not clearly explained, then this method would not be solving the problem of class-incremental learning. However, in the experiments, the method is compared against baselines that are class-incremental methods, which makes the comparison unfair.

---

> ### Author Response · Authors · 2024-12-03
> **Response to the reviewer [concern 1-4, 6]**
>
> **Concern 1: The writing in this paper is poor, making it difficult to follow.**
>
> **Concern 6: The writing in this article is difficult to follow. If the author can improve the writing and satisfactorily address the issues I am concerned about, I would consider raising the score.**
>
> Thank you for your insightful and valuable suggestions. We greatly appreciate your input, as it is crucial for improving the readability of our paper and helping readers better understand our key contributions.
>
> We have reorganized the content of the paper and made several revisions to present our research motivation and primary contributions more clearly. All changes are highlighted in yellow in the PDF.
>
> **Concern 2: The section "The extraction and update of principal directions" in Section 3.3.1 is essentially the GPM (ICLR 2021) [1] algorithm and should not be considered a contribution of this work. It should not be described in such detail.**
>
> **Concern 3: It is recommended that the author include an overview section at the beginning of Section 3 to introduce the whole process of the proposed method, and at the end of Section 3, present Algorithm to summarize the whole process of the method.**
>
> **Concern 4: There is a lack of a problem formulation section that explains what problem you are solving and what settings you are considering. Specifically, existing continual learning settings include task incremental learning, class incremental learning, domain incremental learning, and so on. This paper does not even mention what continual learning setting the authors are considering.**
>
> We have streamlined the content of the section "The extraction and update of principal directions" from the original Section 3.3.1, now Section 4.3.1, and moved less relevant parts to the appendix of the revised PDF. Additionally, we have added an overview at the beginning of the original Section 3, now Section 4, to introduce the entire process of the proposed method, explain the problem being addressed, and describe the settings considered in this work.
>
> Finally, we will insert a block of pseudocode at the end of the original Section 3, now Section 4, to summarize the entire process of the method.
>
> **for data in Task i=1....n:**
>
> **do  $Emd_{\theta, i} =Compute Transfer Distance(data)$**
>
> **for j=i...$n_{learned}$:**
>
> **$TD_{\theta,i,j} = \sum_{k=1}^{d} emd_{\theta,i,k} \cdot emd_{\theta,j,k}$**
>
> **t=max($TD_{\theta,i,j}$)**
>
> **if  $TD_{\theta,i,t}$> $TD_{theld}$:**
>
> **Sharing parameters with Task t by DGU**
>
> **else:**
>
> **Add a new branch and select the old branches**

---

> ### Author Response · Authors · 2024-12-03
> **Response to the reviewer [concern 5] and [question 1]**
>
> **Concern 5: The experimental setup is not clearly described, such as learning rate, batch size, epoch, etc., are not provided. The hyperparameter settings for the method are also not clearly explained."
>
> Thank you for your insightful feedback. We added more details about the experimental setup to the revised PDF's main text and Appendix. This setup is listed in the following table.
>
> | Experiment| Learning Rate| Batch Size|Epoch|$TD_{theld}|
> |:-------:|:-------:|:---:|:---:|:---:|
> |CIFAR B0 Inc5|0.0005|64|20|2.5*e-6|
> |CUB B0 Inc5|0.005|128|5|1*e-5|
> |IN-R B0 Inc5|0.0005|64|20|2*e-6|
> |IN-A B0 Inc20|0.0005|64|5|2*e-6|
> |ObjNet B0 Inc5|0.0005|64|20|3*e-4|
> |OmniBench B0 Inc30|0.001|128|20|2*e-5|
> |VTAB B0 Inc10|0.0005|32|40|2*e-3|
>
> **Question 1: The authors only introduced the use of the MoE (Mixture of Experts) structure in Section 3.1, and from Figure 2, it can be seen that the proposed method maintains a task-specific router for each task. Since the router is task-specific, I would like to ask if the proposed method is addressing the class incremental learning problem. If so, since class incremental learning requires task labels are unavailable during the inference phase, how do the proposed method selects the task routers to use during inference for a given test sample? If this is not clearly explained, then this method would not be solving the problem of class-incremental learning. However, in the experiments, the method is compared against baselines that are class-incremental methods, which makes the comparison unfair.**
>
> Thank you for your suggestion! We add how our proposed method selects the task routers in Section 4.1 of the revised PDF.  Since the task ID is not provided during inference, we learn a class center for each task during training. Then, during inference, we calculate the closest class center to the sample and use it to select the corresponding router.

---

### Official Review · Reviewer_ikh8 · 2024-10-31

**Soundness:** 2
**Presentation:** 2
**Contribution:** 1
**Rating:** 3
**Confidence:** 5

**Summary:**

This paper tackles the continual learning problem with pre-trained ViT. The topic is important to the machine learning field. The authors adopt the combination of MOE and GEM to tackle this problem. The proposed method is evaluated on several datasets against other baselines.

**Strengths:**

1.	This paper tackles the continual learning problem with pre-trained ViT. The topic is important to the machine learning field.
2.	The authors adopt the combination of MOE and GEM to tackle this problem.
3.	The proposed method is evaluated on several datasets against other baselines.

**Weaknesses:**

1.	The writing of this paper is problematic, with many overlapping with existing works. For example, the authors have put much effort into the fisher information matrix, which is a common technique in EWC [1], and the bound in Line 228 is also directly copied from [2]. I would suggest the authors improve their writing by highlighting their own contributions and avoiding using vague descriptions.
2.	Overall, this paper makes a minor combination of GEM [3] and MOE-Adapters [4]. The basic idea for gradient projection is directly borrowed from [3], while the way the authors adopt the MOE and LORA is a simple modification of [4] (the only difference lies in the parameter-efficient tuning structure on LORA against Adapter). Hence, I am curious about the contribution of this paper.
3.	The experimental results are also weak, lacking many ablation results. For example, is the method sensitive to hyper-parameters? How do we decide the number of experts? How about removing different modules in the current framework? Are all of them efficient in the whole framework? With these details missing, the current experimental part only shows numerical results against other baselines, which is less informative.

[1] Overcoming catastrophic forgetting in neural networks. PNAS 2017

[2] Coscl: Cooperation of small continual learners is stronger than a big one. CVPR 2022

[3] Gradient Episodic Memory for Continual Learning. NIPS 2017

[4] Boosting continual learning of vision-language models via mixture-of-experts adapters. CVPR 2024

**Questions:**

See Weaknesses

---

> ### Author Response · Authors · 2024-12-03
> **Response to the reviewer [concern 1]**
>
> **Concern 1: The writing of this paper is problematic, with many overlapping with existing works. For example, the authors have put much effort into the fisher information matrix, which is a common technique in EWC [1], and the bound in Line 228 is also directly copied from [2]. I would suggest the authors improve their writing by highlighting their own contributions and avoiding using vague descriptions.**
>
> Thank you for your suggestion for our writing! It is crucial for improving the readability of our paper and helping readers better understand our key contributions. We have reorganized the content of the paper and made several revisions to present our research motivation and primary contributions more clearly.  All changes are highlighted in yellow in the PDF.
>
> 1. **Reorganized content.** In the revised PDF, first, we rewrote the first two paragraphs in Section 1 and incorporated the theoretical analysis of prior works from the appendix into Section 2. Second, we moved some details less relevant to our contributions from Section 4 to the appendix.
> 2. **Writing correction.** We performed a thorough check for spelling, grammar, and figure/table reference numbers and enlarged the text in the main figure for better clarity.

---

> ### Author Response · Authors · 2024-12-03
> **Response to the reviewer [concern 2]**
>
> **Concern 2: Overall, this paper makes a minor combination of GEM [3] and MOE-Adapters [4]. The basic idea for gradient projection is directly borrowed from [3], while the way the authors adopt the MOE and LORA is a simple modification of [4] (the only difference lies in the parameter-efficient tuning structure on LORA against Adapter). Hence, I am curious about the contribution of this paper.**
>
> We respectfully disagree with the suggestion that this paper makes a minor combination of GEM and MOE adapters, and the basic idea for gradient projection is directly borrowed from GEM. GEM works by storing a small subset of previous task data (referred to as episodic memory) and using it to ensure that the model does not significantly degrade its performance on previous tasks when learning a new task. GEM has two key features, which are old sample replay and gradient projection. In Section 2, we analyze prior work and provide a theoretical discussion of these two key features, highlighting their current limitations.
>
> **Differences.** Our method does not rely on old sample replay. For gradient projection, we only apply it to shared experts when the model detects a significant transfer between the current and learned tasks. We have also made substantial improvements to the original gradient projection technique. The original gradient projection method cannot distinguish the degree of transfer and interference between old and new tasks. It restricts gradient updates in all orthogonal directions of old task gradients. When the new task's gradient update direction has opposing components along the old task's gradient direction, the original method severely limits learning, causing forward interference. In contrast, our gradient projection technique applies only when the gradient directions between new and old tasks are orthogonal or share aligned components. Furthermore, based on the detected degree of transfer, we reduce excessive restrictions on gradients with aligned directions. This approach not only helps learn the new task but also improves performance on the old task, enhancing overall model generalization. For more details, refer to Section 4.3.
> As for the combined use of MOE and LoRA, it serves as the structural foundation of our algorithm but is not the main innovation. We have moved some details not directly related to our contributions from Section 4.1 to the appendix in the revised PDF.
>
> Below, I will briefly outline the structure, research motivation, and key contributions of the revised paper.
>
> 1. **Research motivation.** The first paragraph of Section 1 clearly defines the essence of continual learning as maximizing transfer and minimizing interference, while introducing four types of transfer and interference. In the second paragraph, we present our research motivation, highlighting that most continual learning methods fail to fully maximize transfer and minimize interference due to a lack of a fundamental understanding of transfer and interference.
> 2. **Key contributions.** Section 2 provides a more detailed theoretical analysis of prior works, explaining the shortcomings of current methods and reinforcing our research motivation. Section 3 addresses these shortcomings by defining the conditions for transfer and interference in continual learning from the perspective of local optimization. Section 4 introduces a metric for quantifying the degree of transfer and interference across tasks in different model modules and outlines corresponding strategies for handling varying levels of transfer and interference.
> In Section 5, we demonstrate the effectiveness of our approach through experiments on continual learning, transfer, interference, and ablation studies.
>
> Our work builds on the essence of continual learning, providing specific conditions for transfer and interference from the local optimization perspective. Based on these conditions, we propose metrics for measuring transfer and interference and strategies for managing them, to maximize transfer and minimize interference during learning.

---

> ### Author Response · Authors · 2024-12-03
> **Response to the reviewer [concern 3]**
>
> **Concern 3: The experimental results are also weak, lacking many ablation results. For example, is the method sensitive to hyper-parameters? How do we decide the number of experts? How about removing different modules in the current framework? Are all of them efficient in the whole framework? With these details missing, the current experimental part only shows numerical results against other baselines, which is less informative.**
>
> Thank you for your suggestion! We have done and analyzed ablation studies about the effects of different components in Section 5.3 with Fig 1(b) and (c). The ablation experiment results report the incremental performance and parameter efficiency of different variations on the VTAB dataset. This demonstrates that the combination of all components is necessary to achieve a balance between accuracy and model parameters.
>
> As described in Section 5.2 of the paper, we conducted baseline experiments on continual learning using the ViT-B/16-IN21K and ViT-B/16-IN1K pre-trained models. From the results in Fig. 1 and Table 1, we observe that the choice of the pre-trained model does not affect the superiority of our method compared to other continual learning algorithms.
>
> We selected the appropriate hyperparameters and the number of expert branches through experiments. All details of our hyperparameter selection process will be included in the appendix of the final version of the PDF.

---

### Official Review · Reviewer_QgG7 · 2024-11-03

**Soundness:** 3
**Presentation:** 1
**Contribution:** 2
**Rating:** 5
**Confidence:** 3

**Summary:**

This paper addresses the challenges of continual learning (CL), particularly in maximizing knowledge transfer while minimizing interference among tasks. The authors propose a novel method that enhances the performance of pretrained models by integrating proactive mechanisms to detect transfer and interference at the local optimization level. The framework was tested on several benchmark datasets, demonstrating significant improvements in accuracy compared to traditional CL methods.

**Strengths:**

1. The authors introduce a novel adaptive continual learning strategy that effectively balances transfer and interference.
2. The empirical validation shows promising results across various benchmarks.

**Weaknesses:**

1. While the authors present a comprehensive framework for continual learning, the writing is challenging for readers to follow. In the methods section, many techniques are employed, such as MoE, FIM, and PCA; however, the reasons and intuitions behind using these techniques are only briefly mentioned. Furthermore, the authors do not reference whether these commonly used techniques have been applied in this field, making it difficult to assess the contribution of this paper.
2. Additionally, the use of numerous symbols in the methods section complicates comprehension, as their meanings are not clearly defined. The authors should simplify irrelevant introductions (e.g., the formulation of LoRA, which is introduced but not further used) and consider providing a table listing all used symbols.
3. The claims made in Section 3.2 are weak and would benefit from additional evidence for support.
4. Ablation studies are lacking; the effects of different components (e.g., various pretrained models) should be evaluated.
5. Minor issues:
  - Typically, only the best results should be highlighted in bold, rather than including all results.
  - Typos include a missing space in line 424, potential inconsistency with "k_t" in line 297, and incorrect subscript usage in line 342.

**Questions:**

Why does the formulation of $\beta$ in Eq. (10) seem misaligned with its definition?

---

> ### Author Response · Authors · 2024-12-03
> **Response to the reviewer [concern 1-3,5] and [question 1]**
>
> **Concern 1: While the authors present a comprehensive framework for continual learning, the writing is challenging for readers to follow. In the methods section, many techniques are employed, such as MoE, FIM, and PCA; however, the reasons and intuitions behind using these techniques are only briefly mentioned. Furthermore, the authors do not reference whether these commonly used techniques have been applied in this field, making it difficult to assess the contribution of this paper.**
>
> **Concern 2: Additionally, the use of numerous symbols in the methods section complicates comprehension, as their meanings are not clearly defined. The authors should simplify irrelevant introductions (e.g., the formulation of LoRA, which is introduced but not further used) and consider providing a table listing all used symbols.**
>
> **Concern 3: The claims made in Section 3.2 are weak and would benefit from additional evidence for support.**
>
> **Concern 5: Minor issues:
> Typically, only the best results should be highlighted in bold, rather than including all results.
> Typos include a missing space in line 424, potential inconsistency with "k_t" in line 297, and incorrect subscript usage in line 342.**
>
> **Question 1: Why does the formulation of in Eq. (10) seem misaligned with its definition?**
>
> Thank you for your insightful and valuable suggestions. We greatly appreciate your input, as it is crucial for improving the readability of our paper and helping readers better understand our key contributions.
>
> We have reorganized the content of the paper and made several revisions to present our research motivation and primary contributions more clearly. All changes are highlighted in yellow in the PDF.
>
> 1. **Reorganized content.** In the revised PDF, first, we rewrote the first two paragraphs in Section 1 and incorporated the theoretical analysis of prior works from the appendix into Section 2. Section 2 provides an analysis of the strengths and weaknesses of prior work, highlighting the contributions of our approach. Second, we moved some details less relevant to our contributions from Section 4 to the appendix. It has also added more detailed explanations of the reasons behind our method design. In the appendix, we present the theoretical proof of our algorithm's effectiveness.
>
> 2. **Writing correction.** We thoroughly checked spelling, grammar, and figure/table reference numbers and enlarged the text in the main figure for better clarity.

---

> ### Author Response · Authors · 2024-12-03
> **Response to the reviewer [concern 4]**
>
> **Concern 4: Ablation studies are lacking; the effects of different components (e.g., various pretrained models) should be evaluated.**
>
> We have done and analyzed ablation studies about the effects of different components in Section 5.3 with Fig 1(b) and (c). The ablation experiment results report the incremental performance and parameter efficiency of different variations on the VTAB dataset. This demonstrates that the combination of all components is necessary to achieve a balance between accuracy and model parameters.
>
> As described in Section 5.2 of the paper, we conducted baseline experiments on continual learning using the ViT-B/16-IN21K and ViT-B/16-IN1K pre-trained models. From the results in Fig. 1 and Table 1, we observe that the choice of the pre-trained model does not affect the superiority of our method compared to other continual learning algorithms.

---

### Official Review · Reviewer_snTK · 2024-11-04

**Soundness:** 3
**Presentation:** 1
**Contribution:** 2
**Rating:** 5
**Confidence:** 3

**Summary:**

This paper proposes a new continual learning approach that addresses the importance of proactive detection of transfer and interference. They first propose a new metric for task transfer distance measurement, and accordingly design a dynamic parameter update mechanism based on LoRA finetuning on a subset of experts in MOE mechanism, such that the previous and current tasks can be well balanced. Experiments on 7 benchmarks are provided to show the transfer effectiveness of the proposed approach.

**Strengths:**

1. The proactive detection of transfer distance between tasks and a corresponding adaptive continual learning design is generally reasonable and novel to me.

2. Extensive evaluations on multiple benchmark datasets are provided.

**Weaknesses:**

1. The authors make a huge effort to explain the design details of the proposed approach but lack enough discussion on the high-level intuitions and key insights behind their designs. Therefore, it is hard to appreciate the paper's value through reading the paper.

2. Generally, the presentation in the paper is poor and hard to follow. Typos and grammar issues happen frequently. The fonts in the figures are too small. The figure references in Section 4.3 seem to be all wrongly placed.

3. The experiments are mostly about how the proposed approach outperforms the existing approaches, but without in-depth analysis on the reasons behind the results. Besides, the authors address the importance of the transfer and interference, but the analysis presented in Section 4.4 is weak and fails to highlight the value of the distinguishment.

4. The proposed approach only works with the pretrained Transformer architectures, but not other models, but the authors do not clarify that clearly in early sections of the paper.

**Questions:**

1. Is the proposed approach only applicable to Transformer models in vision tasks?

2. In Section 3.3.2, how many and which old branches do you select as "a few old branches" to participate in learning the new task?

---

> ### Author Response · Authors · 2024-12-02
> **Response to the reviewer [concern 1] and [concern 2]**
>
> **Concern 1: The authors make a huge effort to explain the design details of the proposed approach but lack enough discussion on the high-level intuitions and key insights behind their designs. Therefore, it is hard to appreciate the paper's value through reading the paper.**
>
> **Concern 2: Generally, the presentation in the paper is poor and hard to follow. Typos and grammar issues happen frequently. The fonts in the figures are too small. The figure references in Section 4.3 seem to be all wrongly placed.**
>
> Thank you for your insightful and valuable suggestions. We greatly appreciate your input, as it is crucial for improving the readability of our paper and helping readers better understand our key contributions.
>
> We have reorganized the content of the paper and made several revisions to present our research motivation and primary contributions more clearly.  All changes are highlighted in yellow in the PDF.
>
> 1. **Reorganized content.** In the revised PDF, first, we rewrote the first two paragraphs in Section 1 and incorporated the theoretical analysis of prior works from the appendix into Section 2. Second, we moved some details less relevant to our contributions from Section 4 to the appendix.
> 2. **Writing correction.** We performed a thorough check for spelling, grammar, and figure/table reference numbers and enlarged the text in the main figure for better clarity.
>
> Below, I will briefly outline the structure, research motivation, and key contributions of the revised paper.
>
> 1. **Research motivation.** The first paragraph of Section 1 clearly defines the essence of continual learning as maximizing transfer and minimizing interference, while introducing four types of transfer and interference. In the second paragraph, we present our research motivation, highlighting that most continual learning methods fail to fully maximize transfer and minimize interference due to a lack of a fundamental understanding of transfer and interference.
> 2. **Key contributions.** Section 2 provides a more detailed theoretical analysis of prior works, explaining the shortcomings of current methods and reinforcing our research motivation. Section 3 addresses these shortcomings by defining the conditions for transfer and interference in continual learning from the perspective of local optimization. Section 4 introduces a metric for quantifying the degree of transfer and interference across tasks in different model modules and outlines corresponding strategies for handling varying levels of transfer and interference.
> In Section 5, we demonstrate the effectiveness of our approach through experiments on continual learning, transfer, interference, and ablation studies.
>
> Our work builds on the essence of continual learning, providing specific conditions for transfer and interference from the local optimization perspective. Based on these conditions, we propose metrics for measuring transfer and interference and strategies for managing them, to maximize transfer and minimize interference during learning.

---

> ### Author Response · Authors · 2024-12-03
> **Response to the reviewer [concern 3]**
>
> **Concern3: The experiments are mostly about how the proposed approach outperforms the existing approaches, but without in-depth analysis on the reasons behind the results. Besides, the authors address the importance of the transfer and interference, but the analysis presented in Section 4.4 is weak and fails to highlight the value of the distinguishment.**
>
> This feedback is incredibly valuable! To better understand why our proposed method outperforms others, we compared it with the EASE method, which exhibits similar performance, on the VTAB dataset. Specifically, we analyzed how accuracy changes for each task as the number of learned categories increases. By analyzing VTAB, we found that Tasks 1 and 4 both involve satellite remote sensing images, while Tasks 3 and 5 both focus on natural images.
>
> As shown in the Table below, the accuracy of EASE on Task 1 sharply declined when it learned the similar Task 4. In contrast, our method improved accuracy on Task 1 by learning Task 4. A similar trend was also observed between Tasks 3 and 5. Additionally, when comparing the accuracy of both methods on the first learning of task 4, our method outperformed EASE. This demonstrates that our method effectively avoids backward interference and maximizes both forward and backward transfer between similar tasks, resulting in better accuracy. All results and analyses will be included in the appendix of the final version.
> | Task| method| 10|20|30|40|50|
> |:-------:|:-------:|:---:|:---:|:---:|:---:|:---:|
> |Task 1|EASE|95.5|95.2|95.1|**93.4**|93.1|
> |           |Ours|95.5|95.2|95.2|**95.5**|95.2|
> |Task 2|EASE||84.3|84.3|84.3|84.3|
> |           |Ours||84.3|84.3|84.3|84.3|
> |Task 3|EASE|||**94.8**|94.8|94.8|
> |           |Ours|||**95.4**|95.3|95.6|
> |Task 4|EASE||||**92.7**|92.7|
> |           |Ours||||**93.6**|93.4|
> |Task 5|EASE|||||100|
> |           |Ours|||||100|
>
> Additionally, we have moved the table recording the number of adapters learned in different model blocks from the Appendix to Section 5.4. Table 3 in the revised PDF shows that, in our method, certain model modules share parameters. For instance, in the VTAB, Task 1 and Task 4 share parameters, as do Task 3 and Task 5. The table also reveals a clear hierarchical pattern of parameter sharing. In some cases, earlier blocks exhibit higher-level sharing, indicating similar low-level features, while later blocks show more sharing, reflecting similar high-level features. This demonstrates how our algorithm effectively balances accuracy with model parameter efficiency.

---

> ### Author Response · Authors · 2024-12-03
> **Response to other concerns and questions**
>
> **Concern 4: The proposed approach only works with the pretrained Transformer architectures, but not other models, but the authors do not clarify that clearly in early sections of the paper.**
>
> **Question 1: Is the proposed approach only applicable to Transformer models in vision tasks?**
>
> Thank you for your question. The method proposed in this paper is built on a pre-trained Transformer architecture. To clarify this assumption, we have added a specific description at the beginning of Section 4 in the revised PDF. Our algorithm demonstrates that the transfer distance metric, based on the Fisher matrix and gradient directions, along with the corresponding continual learning methods for different transfer distances, relies on gradient updates and model architecture. Thus, our method is theoretically applicable to all gradient-based models. Due to time constraints during the rebuttal period, we will include experiments with other model architectures, such as ResNet, in the appendix of the final paper version.
>
> **Question 2: In Section 3.3.2, how many and which old branches do you select as "a few old branches" to participate in learning the new task?**
>
> We chose 1 old branch as "a few old branches" to participate in learning the new task, and every task has at least 2 branches. We have added more details on hyperparameter selection in the Appendix of the revised PDF.

---

### Meta-Review · Area_Chair_9MbM · 2024-12-22

**Metareview:**

The paper proposes an adaptive continual learning strategy that proactively detects transfer and interference. It uses the Fisher matrix and gradient update directions to derive conditions for identifying all types of transfer and interference from the perspective of parameter sharing and optimization techniques. Experimental results on seven benchmarks demonstrate the effectiveness of the proposed method.

**Strengths**

- Detecting transfer and interference is a critical aspect of continual learning.
- The use of the Fisher matrix and gradient update directions provides a meaningful way to understand the connection between model parameters and transfer/interference in continual learning.

**Weaknesses**

- The paper lacks clarity in its justification for the choice of existing techniques and does not adequately explain how these techniques are integrated in a novel and nontrivial manner to advance the state-of-the-art in continual learning.
- The experimental results are insufficient, with key ablation studies missing, which limits the ability to fully assess the contributions of the proposed method.

**Overall Assessment**
The paper requires significant revision to better articulate its technical contributions. The authors are suggested to provide stronger theoretical insights and a more comprehensive empirical evaluation, including key ablation studies, to clearly establish the impact and novelty of their approach.

**Additional Comments On Reviewer Discussion:**

The authors’ response addressed some of the concerns raised during the review process; however, several major issues remain inadequately resolved. The authors are encouraged to carefully consider the reviewers’ suggestions to further refine and improve their work for a future submission.

---

### Decision · Program_Chairs · 2025-01-22

Reject